# Prototypical VoteNet for Few-Shot 3D Point Cloud Object Detection

**Shizhen Zhao, Xiaojuan Qi***
The University of Hong Kong
{zhaosz,xjqi}@eee.hku.hk

## Abstract

Most existing 3D point cloud object detection approaches heavily rely on large amounts of labeled training data. However, the labeling process is costly and time-consuming. This paper considers few-shot 3D point cloud object detection, where only a few annotated samples of novel classes are needed with abundant samples of base classes. To this end, we propose Prototypical VoteNet to recognize and localize novel instances, which incorporates two new modules: Prototypical Vote Module (PVM) and Prototypical Head Module (PHM). Specifically, as the 3D basic geometric structures can be shared among categories, PVM is designed to leverage class-agnostic geometric prototypes, which are learned from base classes, to refine local features of novel categories. Then PHM is proposed to utilize class prototypes to enhance the global feature of each object, facilitating subsequent object localization and classification, which is trained by the episodic training strategy. To evaluate the model in this new setting, we contribute two new benchmark datasets, FS-ScanNet and FS-SUNRGBD. We conduct extensive experiments to demonstrate the effectiveness of Prototypical VoteNet, and our proposed method shows significant and consistent improvements compared to baselines on two benchmark datasets. This project will be available at `https://shizhen-zhao.github.io/FS3D_page/`.

## 1 Introduction

3D object detection aims to localize and recognize objects from point clouds with many applications in augmented reality, autonomous driving, and robotics manipulation. Recently, a number of fully supervised 3D object detection approaches have made remarkable progress with deep learning [22, 18, 31, 24]. Nonetheless, their success heavily relies on large amounts of labeled training data, which are time-consuming and costly to obtain. On the contrary, a human can quickly learn to recognize novel classes by seeing only a few samples. To imitate such human ability, we consider few-shot point cloud 3D object detection, which aims to train a model to recognize novel categorizes from limited annotated samples of novel classes together with sufficient annotated data of base classes.

Few-shot learning has been extensively studied in various 2D visual understanding tasks such as object detection [39, 40, 43, 45], image classification [14, 9, 3, 32], and semantic segmentation [23, 21, 48, 19]. Early attempts [9, 16, 11, 38] employ meta-learning to learn transferable knowledge from a collection of tasks and attained remarkable progress. Recently, benefited from large-scale datasets (*e.g.* ImageNet [6]) and advanced pre-training methods [27, 49, 10, 54], finetuning large-scale pre-trained visual models on down-stream few-shot datasets emerges as an effective approach to address this problem [33, 39, 55]. Among different streams of work, prototype-based methods [42, 53, 20, 17] have been incorporated into both streams and show the great advantages, since they can capture the

---

*Corresponding author

36th Conference on Neural Information Processing Systems (NeurIPS 2022).

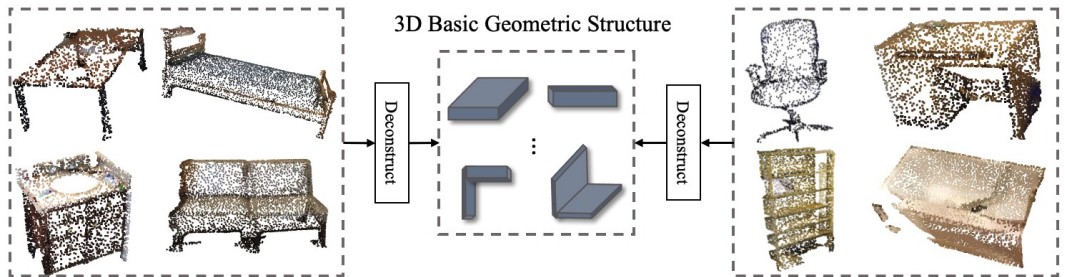

Figure 1: Illustration of the basic geometry of 3D objects, which can be shared among classes.

representative features of categories that can be further utilized for feature refinement [45, 51] or classification [26, 32].

This motivates us to explore effective 3D cues to build prototypes for few-shot 3D detection. Different from 2D visual data, 3D data can get rid of distortions caused by perspective projections, and offer geometric cues with accurate shape and scale information. Besides, 3D primitives to constitute objects can often be shared among different categories. For instance, as shown in Figure 1, rectangular plates and corners can be found in many categories. Based on these observations, in this work, we propose Prototypical VoteNet, which employs such robust 3D shape and primitive clues to design geometric prototypes to facilitate representation learning in the few-shot setting.

Prototypical VoteNet incorporates two new modules, namely Prototypical Vote Module (PVM) and Prototypical Head Module (PHM), to enhance local and global feature learning, respectively, for few-shot 3D detection. Specifically, based on extracted features from a backbone network (*i.e.* PointNet++ [25]), PVM firstly constructs a class-agnostic 3D primitive memory bank to store geometric prototypes, which are shared by all categories and updated iteratively during training. To exploit the transferability of geometric structures, PVM then incorporates a multi-head cross-attention module to associate geometric prototypes with points in a given scene and utilize them to refine their feature representations. PVM is majorly developed to exploit shared geometric structures among base and novel categories to enhance feature learning of local information in the few-shot setting. Further, to facilitate learning discriminative features for object categorization, PHM is designed to employ a multi-head cross-attention module and leverage class-specific prototypes from a few support samples to refine global representations of objects. Moreover, episodic training [32, 38] is adopted to simulate few-shot circumstances, where PHM is trained by a distribution of similar few-shot tasks instead of only one target object detection task.

Our **contributions** are listed as follows:

- We are the first to study the promising few-shot 3D point cloud object detection task, which allows a model to detect new classes, given a few examples.

- We propose Prototypical VoteNet, which incorporates Prototypical Vote Module and Prototypical Head Module, to address this new challenge. Prototypical Vote Module leverages class-agnostic geometric prototypes to enhance the local features of novel samples. Prototypical Head Module utilizes the class-specific prototypes to refine the object features with the aid of episode training.

- We contribute two new benchmark dataset settings called FS-ScanNet and FS-SUNRGBD, which are specifically designed for this problem. Our experimental results on these two benchmark datasets show that the proposed model effectively addresses the few-shot 3D point cloud object detection problem, yielding significant improvement over several competitive baseline approaches.

## 2 Related Work

**3D Point Cloud Object Detection.** Current 3D point cloud object detection approaches can be divided into two streams: Grid Projection/Voxelization based [46, 15, 35, 4, 52, 41] and point-based [30, 22, 18, 8, 2]. The former projects point cloud to 2D grids or 3D voxels so that the advanced convolutional networks can be directly applied. The latter methods take the raw point cloud

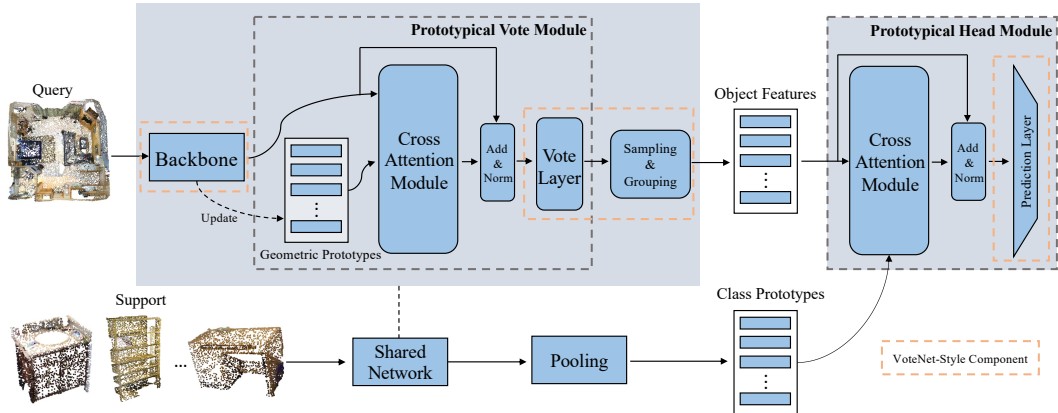

Figure 2: Illustration of Prototypical VoteNet. Prototypical VoteNet introduces two modules for few-shot 3D detection: 1) Prototypical Vote Module for enhancing local feature representation of novel samples by leveraging the geometric prototypes, 2) Prototypical Head Module for refining global features of novel objects, utilizing the class-specific prototypes.

feature extraction network such as PointNet++ [25] to generate point-wise features for the subsequent detection. Although these fully supervised approaches achieved promising 3D detection performance, their requirement for large amounts of training data precludes their application in many real-world scenarios where training data is costly or hard to acquire. To alleviate this limitation, we explore the direction of few-shot 3D object detection in this paper.

**Few-Shot Recognition.** Few-shot recognition aims to classify novel instances with abundant base samples and a few novel samples. Simple pre-training and finetuning approaches first train the model on the base classes, then finetune the model on the novel categories [3, 7]. Meta-learning based methods [9, 16, 11, 38, 32] are proposed to learn classifier across tasks and then transfer to the few-shot classification task. The most related work is Prototypical Network [32], which represents a class as one prototype so that classification can be performed by computing distances to the prototype representation of each class. The above works mainly focus on 2D image understanding. Recently, some few-shot learning approaches for point cloud understanding [29, 51, 47] are proposed. For instance, Sharma *et al.* [51] propose a graph-based method to propagate the knowledge from few-shot samples to the input point cloud. However, there is no work studying few-shot 3D point cloud object detection. In this paper, we first study this problem and introduce the spirit of Prototypical Network into few-shot 3D object detection with 3D geometric prototypes and 3D class-specific prototypes.

**2D Few-shot Object Detection.** Most existing 2D few-shot detectors employ a meta-learning [40, 14, 45] or fine-tuning based mechanism [44, 43, 26, 36]. Particularly, Kang *et al.* [14] propose a one-stage few-shot detector which contains a meta feature learner and a feature re-weighting module. Meta R-CNN [45] presents meta-learning over RoI (Region-of-Interest) features and incorporates it into Faster R-CNN [28] and Mask R-CNN [11]. TFA [39] reveals that simply fine-tuning the box classifier and regressor outperforms many meta-learning based methods. Cao *et al.* [1] improve the few-shot detection performance by associating each novel class with a well-trained base class based on their semantic similarity.

## 3 Our Approach

In few-shot 3D point cloud object detection (FS3D), the object class set $\mathbb{C}$ is split into $\mathbb{C}_{\text{base}}$ and $\mathbb{C}_{\text{novel}}$ such that $\mathbb{C} = \mathbb{C}_{\text{base}} \cup \mathbb{C}_{\text{novel}}$ and $\mathbb{C}_{\text{base}} \cap \mathbb{C}_{\text{novel}} = \emptyset$. For each class $r \in \mathbb{C}$, its annotation dataset $T_r$ contains all the data samples with object bounding boxes, that is $T_r = \{(u, P) | u \in \mathbb{R}^6, P \in \mathbb{R}^{N \times 3}\}$. Here, $(u, P)$ is a 3D object bounding box $u = (x, y, z, h, w, l)$, representing box center locations and box dimensions, in a point cloud scene $P$.

There are only a few examples/shots for each novel class $r \in \mathbb{C}_{\text{novel}}$, which are known as support samples. Besides, there are plenty of annotated samples for each base class $r \in \mathbb{C}_{\text{base}}$. Given the above dataset, FS3D aims to train a model to detect object instances in the novel classes leveraging

such sufficient annotations for base categories $\mathbb{C}_{\text{base}}$ and limited annotations for novel categories $\mathbb{C}_{\text{novel}}$.

In the following, we introduce Prototypical VoteNet for few-shot 3D object detection. We will describe the preliminaries of our framework in Section 3.1, which adopts the architecture of VoteNet-style 3D detectors [24, 50, 2]. Then, we present Prototypical VoteNet consisting of Prototypical Vote Module (Section 3.2.1) and Prototypical Head Module (Section 3.2.2) to enhance feature learning for FS3D.

## 3.1 Preliminaries

VoteNet-style 3D detectors [24, 50, 2] takes a point cloud scene $P_i$ as input, and localizes and categorizes 3D objects. As shown in Figure 2, it firstly incorporates a 3D backbone network (*i.e.* PointNet++ [25]) parameterized by $\theta_1$ with downsampling layers for point feature extraction as Equation (1).

$$F_i = h_1(P_i; \theta_1), \tag{1}$$

where $N$ and $M$ represent the original and subsampled number of points, respectively, $P_i \in \mathbb{R}^{N \times 3}$ represents an input point cloud scene $i$, and $F_i \in \mathbb{R}^{M \times (3+d)}$ is the subsampled scene points (also called seeds) with $d$-dimensional features and 3-dimensional location coordinates.

Then, $F_i$ is fed into the vote module with parameters $\theta_2$ which outputs a 3-dimensional coordinate offset $\Delta d_j = (\Delta x_j, \Delta y_j, \Delta z_j)$ relative to its corresponding object center $c = (c_x, c_y, c_z)$ and a residual feature vector $\Delta f_j$ for each point $j$ in $F_i = \{f_j\}_i$ as in Equation (2).

$$\{\Delta d_j, \Delta f_j\}_i = h_2(F_i; \theta_2). \tag{2}$$

Given the predicted offset $\Delta d_j$, the estimated corresponding object center $c_j = (c_{x_j}, c_{y_j}, c_{z_j})$ that point $j$ belongs to can be calculated as Equation (3).

$$c_{x_j} = x_j + \Delta x_j, c_{y_j} = y_j + \Delta y_j, c_{z_j} = z_j + \Delta z_j. \tag{3}$$

Similarly, the point features are updated as $F_i \leftarrow F_i + \Delta F_i$ where $\Delta F_i = \{\Delta f_j\}_i$.

Next, the detector samples object centers from $\{(c_{x_j}, c_{y_j}, c_{z_j})\}_i$ using farthest point sampling and group points with nearby centers together (see Figure 2: Sampling & grouping) to form a set of object proposals $O_i = \{o_t\}_i$. Each object proposal is characterized by a feature vector $f_{o_t}$ which is obtained by applying a max pooling operation on features of all points belonging to $o_t$.

Further, equipped with object features $\{f_{o_t}\}_i$, the prediction layer with parameters $\theta_3$ is adopted to yield the bounding boxes $b_t$, objectiveness scores $s_t$, and classification logits $r_t$ for each object proposal $o_t$ following Equation (4).

$$\{b_t, s_t, r_t\}_i = h_3(\{f_{o_t}\}_i; \theta_3). \tag{4}$$

## 3.2 Prototypical VoteNet

Here, we present Prototypical VoteNet which incorporates two new designs – Prototypical Vote Module (PVM) and Prototypical Head Module (PHM) to improve feature learning for novel categories with few annotated samples (see Figure 2). Specifically, PVM builds a class-agnostic memory bank of geometric prototypes $\mathcal{G} = \{g_k\}_{k=1}^K$ with a size of $K$, which models transferable class-agnostic 3D primitives learned from rich base categories, and further employs them to enhance local feature representation for novel categories via a multi-head cross-attention module. The enhanced features are then utilized by the Vote Layer to output the offset of coordinates and features as Equation (2). Second, to facilitate learning discriminative features for novel class prediction, PHM employs an attention-based design to leverage class-specific prototypes $\mathcal{E} = \{e_r\}_{r=1}^R$ extracted from the support set $D_{\text{support}}$ with $R$ categories to refine global discriminate feature for representing each object proposal (see Figure 2). The output features are fed to the prediction layer for producing results as Equation (4). To make the model more generalizable to novel classes, we exploit the episodic training [32, 38] strategy to train PHM, where a distribution of similar few-shot tasks instead of only one object detection task is learned in the training phase. PVM and PHM are elaborated in the following sections.

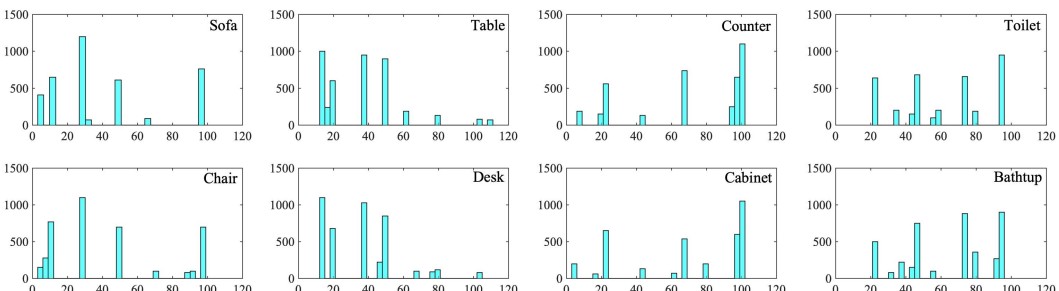

Figure 3: Visualization of the assignment of object point features to the geometric prototypes, where the horizontal axis represents the index of geometric prototypes and the vertical axis represents the number of assignments.

### 3.2.1 Prototypical Vote Module

Given input features $F_i$ extracted by a backbone network, Prototypical Vote Module constructs class-agnostic geometric prototypes $\mathcal{G} = \{g_k\}_{k=1}^K$ and then uses them to enhance local point features with an attention module.

**Geometric Prototype Construction.** At the beginning, $\mathcal{G} = \{g_k\}_{k=1}^K \in \mathbb{R}^{d \times K}$ is randomly initialized. During training, $\mathcal{G}$ is iteratively updated with a momentum based on point features of foreground objects. Specifically, for each update, given $\mathcal{G} = \{g_k\}_{k=1}^K$ and all the foreground points $\{p_m\}_{m=1}^{M_f}$ with features $\{f_m\}_{m=1}^{M_f}$, where $M_f$ is the number of foreground points in the current batch, we assign each point to its nearest geometric prototype based on feature space distance. Then, for each prototype $g_k$, we have a group of points $\{p_m\}_k$ with features represented as $\{f_m\}_k$ assigned to it. Point features in one group are averaged to update the corresponding geometric prototype as Equation (5).

$$g_k \leftarrow \gamma * g_k + (1 - \gamma)\overline{f}_k, \text{where } \overline{f}_k = \text{average}(\{f_m\}_k). \tag{5}$$

Here $\gamma \in [0, 1]$ is the momentum coefficient for updating geometric prototypes in a momentum manner, serving as a moving average over all training samples. Since one point feature is related to one geometric prototype, we call this one-hot assignment strategy as hard assignment. An alternative to the hard assignment is the soft assignment, which calculates the similarity between a point features with all geometric prototypes. Empirically, we found that hard assignment results in more effective grouping versus soft assignment. More details can be found in the supplementary material.

**Geometric Prototypical Guided Local Feature Enrichment.** Given the geometric prototypes $\mathcal{G} = \{g_k\}_{k=1}^K$ and point features $F_i = \{f_j\}_i$ of a scene $i$, PVM further employs a multi-head cross-attention module [37] to refine the point features. Specifically, the multi-head attention network uses the point features $F_i = \{f_j\}_i$ as query, geometric prototypes $\mathcal{G} = \{g_k\}_{k=1}^K$ as key and value where linear transformation functions with weights represented as $Q_h, U_h, V_h$ are applied to encode query, key and value respectively. Here, $h$ represents the head index. Then, for each head $h$, the query point feature is updated by softly aggregating the value features where the soft aggregation weight is determined by the similarity between the query point feature and corresponding key feature. The final point feature $f_j$ is updated by summing over outputs from all heads as Equation (6).

$$f_j \leftarrow \text{Cross\_Att}(f_j, \{g_k\}) = \sum_{h=1}^H W_h(\sum_{k=1}^K A_{j,k}^h \cdot V_h g_k), \text{where } A_{j,k}^h = \frac{\exp[(Q_h f_j)^T (U_h g_k)]}{\sum_{k=1}^K \exp[(Q_h f_j)^T (U_h g_k)]}. \tag{6}$$

Here, $A_{j,k}^h$ is the soft aggregation weight considering the similarity between the $j$-th query point feature and the $k$-th key feature and used to weight the $k$-th value feature. Through this process, the point feature is refined using geometric prototypes in a weighted manner where prototypes similar to the query point feature will have higher attention weights. This mechanism transfers geometric prototypes learned from base categories with abundant data to model novel points. The multi-head design enables the model to seek similarity measurements from different angles in a learnable manner

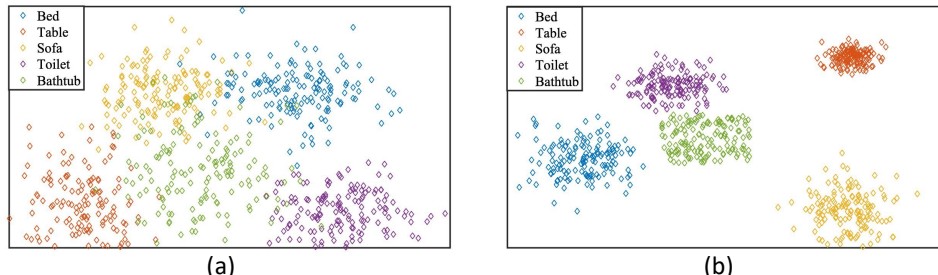

(a)                                                    (b)

Figure 4: t-SNE visualization of the effect of PHM. (a) shows the features without being processed by PHM. (b) shows the features processed by PHM using class-specific prototypes.

to improve robustness. Additionally, in both PHM and PVM, the multi-head attention layer are combined with feed forward FC layers. After refining point features $\{f_j\}_i$, PVM predicts the point offset and residual feature vector $\{\Delta d_j, \Delta F_j\}_i$ as stated in Equation (2). $\Delta d_j$ is explicitly supervised by a regression loss $\mathcal{L}_{\text{vote}}$ used in [25].

**What do the geometric prototypes store?**    To shed more insights on what the geometric prototypes represent, we visualize the frequency of geometry prototypes in different categories using the "assignment". The number of "assignment"'s of object point features to the geometric prototypes is shown in Figure 3, where a point is assigned to the geometric prototype with the highest similarity. In each histogram, the horizontal axis represents the index of geometric prototypes and the vertical axis represents the number of assignments. Note that the first row is the novel classes and the second row is the base classes. Figure 3 shows that two visually similar categories have a similar assignment histogram since they share the basic geometric structures. This indicates that the memory bank of geometric prototypes successfully learns the 3D basic geometric knowledge, which can be a bridge to transfer the geometric knowledge from base classes to novel ones.

### 3.2.2   Prototypical Head Module

As shown in Figure 2, given object proposals $O_i = \{o_t\}_i$ with features $\{f_{o_t}\}_i$ from Sampling & Grouping module, PHM module leverages class-specific prototypes $\{e_r\}$ to refine the object features $f_{o_t}$ for the subsequent object classification and localization. Moreover, for better generalizing to novel categories, PHM is trained by the episodic training scheme, where PHM learns a large number of similar few-shot tasks instead of only one task. Considering the function of PHM, we construct the few-shot tasks that, in each iteration, PHM refines the object features $\{f_{o_t}\}_i$ with the aid of class-specific prototypes, which are extracted from randomly sampled support samples.

In each few-shot task, class-specific prototypes are built based on support sets that are randomly sampled. For class $r$, the class-specific prototype $e_r$ is obtained by averaging the instance features for all support samples in class $r$. The instance feature is derived by applying a max pooling operation over the features of all points belonging to that instance. As shown in Figure 2, with class-specific prototypes $\mathcal{E} = \{e_r\}_{r=1}^R$ for a total of $R$ categories, PHM further employs a multi-head cross-attention module to refine object features $\{f_{o_t}\}_i$. Here, the object features $\{f_{o_t}\}_i$ serve as the query features, class-specific prototypes are used to build value features and key features similar as what has been described in Section 3.2.1. Then, the representation $f_{o_t}$ of each proposal $o_t$ is refined using the outputs of the multi-head attention module, which are weighted sum over the value features and the weight is proportionally to the similarity between the query feature and corresponding key features. This process can be formulated as Equation (7), which is similar to Equation (6).

$$f_{o_t} \leftarrow \text{Cross\_Att}(f_{o_t}, \{e_r\}). \tag{7}$$

Until now, $f_{o_t}$ is refined using class-specific prototypes, injecting class-specific features from given support samples into object-level features. Finally, the refined object features $\{f_{o_t}\}_i$ are fed into the prediction module following Equation (4).

**What does PHM help?**    Figure 4 visualizes the effect of feature refinement by PHM. The experiment is conducted on FS-ScanNet. Figure 4(a) shows the object features, which have not been processed by PHM. Figure 4(b) shows the object features processed by PHM. We could observe that, after

the feature refinement by PHM, the features of each classes become more compact compared to the non-PHM one, which further validates the effectiveness of PHM.

## 3.3 Model Training

The model is trained by the episodic training strategy [32, 38]. The detailed training strategy is included in the supplementary material. We use the standard cross entropy loss $\mathcal{L}_{\text{cls}}$ and smooth-$L_1$ loss [28] $\mathcal{L}_{\text{reg}}$ to supervise the classification and the bounding box regression, respectively. As for the objectness prediction, if a vote is located either within 0.3 meters to a ground truth object center or more than 0.6 meters from any center, it is considered to be positive [24], which is supervised by a cross entropy loss $\mathcal{L}_{\text{obj}}$. Therefore, the overall loss for Prototypical VoteNet is given by,

$$\mathcal{L}_{\text{det}} = \mathcal{L}_{\text{cls}} + \alpha_1 \mathcal{L}_{\text{reg}} + \alpha_2 \mathcal{L}_{\text{obj}} + \alpha_3 \mathcal{L}_{\text{vote}}, \tag{8}$$

where $\alpha_1, \alpha_2, \alpha_3$ is the coefficients to balance the loss contributions.

## 4 Experiments

To our best knowledge, there is no prior study of few-shot point cloud object detection. Therefore, we setup a new benchmark which is described in Section 4.1 & 4.2. Then, we conduct experiments and compare our method with baseline methods in Section 4.3. Third, a series of ablation studies are performed for further analyzing Prototypical VoteNet in Section 4.4. In addition, the implementation details are included in the supplementary material.

### 4.1 Benchmark Setup

**Datasets.** We construct two new benchmark datasets FS-SUNRGBD and FS-ScanNet. Specifically, **FS-SUNRGBD** is derived from SUNRGBD [34]. SUNRGBD consists of 5K RGB-D training images annotated, and the standard evaluation protocol reports performance on 10 categories. We randomly select 4 classes as the novel ones while keeping the remaining ones as the base. In the training set, only $K$ annotated bounding boxes for each novel class are given, where k equals 1, 2, 3, 4 and 5. **FS-ScanNet** is derived from ScanNet [5]. ScanNet consists of 1,513 point clouds, and the annotation of the point clouds corresponds to 18 semantic classes plus one for the unannotated space. Out of its 18 object categories, we randomly select 6 classes as the novel ones, while keeping the remaining as the base. We evaluate with 2 different base/novel splits. In the training set, only $K$ annotated bounding boxes for each novel class are given, where k equals 1, 3 and 5. More details about the new benchmark datasets can be referred to the supplementary material.

**Evaluation Metrics.** We follow the standard evaluation protocol [24] in 3D point cloud object detection by using mean Average Precision(mAP) under different IoU thresholds (*i.e.* 0.25, 0.50), denoted as $\text{AP}_{25}$ and $\text{AP}_{50}$. In addition, in the inference stage, we detect both novel classes and base classes. The performance on base classes is included in the supplementary material.

### 4.2 Benchmark Few-shot 3D Object Detection

We build the first benchmark for few-shot 3D object detection. The benchmark incorporates 4 competitive methods, and three of them are built with few-shot learning strategies, which have been shown to be successful in 2D few-shot object detection.

- Baseline: We abandon PVM and PHM, and train the detector on the base and novel classes together. In this way, it can learn good features from the base classes that are applicable for detecting novel classes.
- VoteNet+TFA [39]: We incorporate a well-designed few-shot object detection method TFA [39] with VoteNet. TFA first trains VoteNet on the training set with abundant samples of base classes. Then only the classifier and the regressor are finetuned with a small balance set containing both base classes and novel classes.
- VoteNet+PT+TFA: The pretraining is proven to be important in 2D few-shot learning, as it learns more generic features, facilitating knowledge transfer from base classes to novel classes. Therefore, we add a pretraining stage, which is borrowed from a self-supervised point cloud contrastive learning method [13], before the training stage of VoteNet+TFA.

| Method | Novel Split 1 | | | | | | Novel Split 2 | | | | | |
|---|---|---|---|---|---|---|---|---|---|---|---|---|
| | 1-shot | | 3-shot | | 5-shot | | 1-shot | | 3-shot | | 5-shot | |
| | $AP_{25}$ | $AP_{50}$ | $AP_{25}$ | $AP_{50}$ | $AP_{25}$ | $AP_{50}$ | $AP_{25}$ | $AP_{50}$ | $AP_{25}$ | $AP_{50}$ | $AP_{25}$ | $AP_{50}$ |
| Baseline | 9.21 | 3.14 | 22.64 | 9.04 | 24.93 | 12.82 | 4.92 | 0.94 | 15.86 | 3.15 | 20.72 | 6.13 |
| VoteNet+TFA | 0.48 | 0.09 | 8.07 | 1.03 | 16.36 | 7.91 | 1.00 | 0.15 | 2.64 | 0.22 | 5.71 | 2.40 |
| VoteNet+PT+TFA | 2.58 | 1.04 | 10.37 | 2.13 | 17.21 | 8.94 | 2.13 | 0.56 | 4.85 | 1.25 | 7.25 | 2.49 |
| Meta VoteNet | 11.01 | 4.20 | 25.73 | 10.99 | 26.68 | 14.40 | 6.06 | 1.01 | 16.93 | 4.51 | 23.83 | 7.17 |
| **Ours** | **15.34** | **8.25** | **31.25** | **16.01** | **32.25** | **19.52** | **11.01** | **2.21** | **21.14** | **8.39** | **28.52** | **12.35** |

Table 1: Results on **FS-ScanNet** using mean Average Precision (mAP) at two different IoU thresholds of 0.25 and 0.50, denoted as $AP_{25}$ and $AP_{50}$.

| Method | 1-shot | | 2-shot | | 3-shot | | 4-shot | | 5-shot | |
|---|---|---|---|---|---|---|---|---|---|---|
| | $AP_{25}$ | $AP_{50}$ | $AP_{25}$ | $AP_{50}$ | $AP_{25}$ | $AP_{50}$ | $AP_{25}$ | $AP_{50}$ | $AP_{25}$ | $AP_{50}$ |
| Baseline | 5.46 | 0.22 | 6.52 | 0.77 | 13.73 | 2.20 | 20.47 | 4.50 | 22.99 | 5.90 |
| VoteNet+TFA | 1.41 | 0.03 | 3.70 | 0.78 | 4.03 | 1.09 | 7.91 | 2.10 | 8.50 | 2.81 |
| VoteNet+PT+TFA | 3.40 | 0.51 | 5.13 | 1.22 | 7.94 | 2.31 | 10.05 | 3.12 | 11.32 | 4.01 |
| Meta VoteNet | 7.04 | 0.98 | 9.23 | 1.34 | 16.24 | 3.12 | 20.10 | 4.69 | 24.41 | 6.05 |
| **Ours** | **12.39** | **1.52** | **14.54** | **3.05** | **21.51** | **6.13** | **24.78** | **7.17** | **29.95** | **8.16** |

Table 2: Results on **FS-SUNRGBD** using mean Average Precision (mAP) at two different IoU thresholds of 0.25 and 0.50, denoted as $AP_{25}$ and $AP_{50}$.

- Meta VoteNet: Meta VoteNet is inspired by Meta RCNN [45]. Meta RCNN develops a meta-learning prediction head, which leverages class prototypes to multiply the RoI features and then predicts if a RoI feature is the category that the class prototype belongs to. In VoteNet, the grouped point features after the vote stage can be treated as the RoI features.

### 4.3 Main Results

**FS-ScanNet.** We first compare the proposed approach with the baselines on FS-ScanNet in Table 1. These results show: 1) VoteNet+TFA has the worst performance. For example, it only achieves 8.07% $AP_{25}$ and 1.03% $AP_{50}$ on 3-shot in Novel Split-1. The reason is that VoteNet+TFA is trained from scratch, and only a few layers are funetuned by novel samples. Therefore, it tends to be overfitting with poor generalization ability on novel classes. 2) Even adding a pretaining stage to VoteNet+TFA, which is termed as VoteNet+PT+TFA, it still performs poorly. For example, it only contributes the improvement of +2.30% $AP_{25}$ and +1.10% $AP_{50}$ to VoteNet+TFA on 3-shot in Novel Split-1. This is different from 2D few-shot object detection. This is potentially caused by the lack of large-scale datasets for pre-training in 3D, while the 2D backbone ResNet [12] has a dataset of over one million images for pre-training. 3) Besides our designed method, Meta VoteNet is superior. For example, Meta VoteNet reaches 25.73% $AP_{25}$ and 16.01% $AP_{50}$ on 3-shot in Novel Split-1. This shows the meta-learning prediction head, derived from Meta RCNN [45], helps to generalize from base classes to novel classes. 4) Our new model Prototypical VoteNet surpasses all competitors by significant margins. For example, Prototypical VoteNet achieves 32.25% $AP_{25}$ and 16.01% $AP_{50}$ on 3-shot in Novel Split-1. This is because our proposed method is equipped with a more generic vote module by learning geometric prototypes, and leverage class prototypes to promote the discriminative feature learning.

**FS-SUNRGBD.** Table 2 shows the result comparison on FS-SUNRGBD. In FS-SUNRGBD, we find that VoteNet+TFA and VoteNet+PT+TFA still have a large gap with other methods due to the overfitting and insufficient pretraining. Meanwhile, Meta VoteNet outperforms other baseline methods. For example, on 3-shot, Meta VoteNet surpasses VoteNet+PT+TFA by +8.34% $AP_{25}$ and +0.81% $AP_{50}$ and achieves 16.24% $AP_{25}$ and 3.12% $AP_{50}$. Finally, our designed Prototypical VoteNet outperforms all the methods. For instance, on 3-shot, Prototypical VoteNet surpasses Meta VoteNet by +5.37% $AP_{25}$ and +3.01% $AP_{50}$, which further validates the effectiveness of our proposed PVM and PHM.

## 4.4 Further Analysis

| Method | 3-shot | | 5-shot | |
|---|---|---|---|---|
| | $AP_{25}$ | $AP_{50}$ | $AP_{25}$ | $AP_{50}$ |
| Baseline | 22.64 | 9.04 | 24.93 | 12.82 |
| +PVM | 27.43 | 13.63 | 28.44 | 16.45 |
| +PHM | 28.76 | 14.04 | 30.13 | 17.51 |
| +PVM+PHM | **31.25** | **16.01** | **32.25** | **19.52** |

Table 3: Ablation study of individual components.

| | Prototype | $AP_{0.25}$ | $AP_{0.50}$ |
|---|---|---|---|
| PVM | Geometric | 31.25 | 16.01 |
| | Self-learning | 28.34 | 14.01 |
| PHM | Class | 31.25 | 16.01 |
| | Self-learning | 27.45 | 13.67 |

Table 4: Ablation study of Prototypes.

**Contributions of individual components.** We conduct ablation studies on 3-shot and 5-shot in split-1 of FS-ScanNet with results shown in Table 3. The results show that both of them are effective on their own. For example, on 3-shot, PVM contributes the improvement of +4.19% $AP_{25}$ and +4.59% $AP_{50}$, and PHM contributes the improvement of +6.12% $AP_{25}$ and +5.00% $AP_{50}$. Moreover, when combined, the best performance, 31.25% $AP_{25}$ and 16.01% $AP_{50}$, is achieved.

**Effectiveness of Prototypes.** Table 4 shows the effectiveness of two kinds of prototypes. In order to validate the geometric prototypes, we displace them by the self-learning ones, which are randomly initialized and updated by the gradient descend during the model training. Table 4 results show that the performance significantly degrades. To validate the effectiveness of class prototypes, we also alter them by the randomly initialized self-learning prototypes. Similarly, the performance drops drastically due to the lack of class prototypes.

| # Prototype | $AP_{0.25}$ | $AP_{0.50}$ |
|---|---|---|
| $K = 30$ | 29.98 | 15.01 |
| $K = 60$ | 30.24 | 15.54 |
| $K = 90$ | 31.10 | 15.98 |
| $K = 120$ | 31.25 | 16.01 |
| $K = 150$ | 31.01 | 15.89 |

Table 5: Ablation study of memory bank size.

| Coefficient $m$ | $AP_{0.25}$ | $AP_{0.50}$ |
|---|---|---|
| $\gamma = 0.2$ | 29.50 | 14.65 |
| $\gamma = 0.9$ | 30.55 | 15.15 |
| $\gamma = 0.99$ | 30.71 | 15.30 |
| $\gamma = 0.999$ | 31.01 | 15.89 |
| $\gamma = 0.9999$ | 30.90 | 15.79 |

Table 6: Ablation study of coefficient $\gamma$.

**Size of Memory Bank.** Table 5 studies the size of the memory bank containing the geometric prototypes. This ablation study is performed on 3-shot in split-1 of FS-ScanNet. The value of K is set to $\{30, 60, 90, 120, 150\}$. For K = 30, the memory bank only contains 30 geometric prototypes, which only achieves 29.98% $AP_{25}$ and 15.01% $AP_{50}$. Furthermore, when using more prototypes (i.e., K = 120), there will be an obvious performance improvement, which reaches 31.25% $AP_{25}$ and 16.01% $AP_{50}$. However, when continuous increasing K, there will be no improvement. Therefore, we set the size of the memory bank to be 120.

**Coefficient $\gamma$.** Table 6 shows the effect of momentum coefficient ($\gamma$ in Equation (5)). The experiment is performed on 3-shot in split 1 of FS-ScanNet. The results show that, when using a relatively large coefficient (i.e., $\gamma \in [0.999, 0.9999]$), the model performs well, compared with the model using a small momentum coefficient (i.e., $\gamma \in [0.9, 0.99]$). Moreover, the performance drops when using a small value of $\gamma = 0.2$. The is potentially because a small momentum coefficient might bring about unstable prototype representation with rapid prototype updating.

| Method | Novel Split 1 | | | | | | Novel Split 2 | | | | | |
|---|---|---|---|---|---|---|---|---|---|---|---|---|
| | 1-shot | | 3-shot | | 5-shot | | 1-shot | | 3-shot | | 5-shot | |
| | $AP_{25}$ | $AP_{50}$ | $AP_{25}$ | $AP_{50}$ | $AP_{25}$ | $AP_{50}$ | $AP_{25}$ | $AP_{50}$ | $AP_{25}$ | $AP_{50}$ | $AP_{25}$ | $AP_{50}$ |
| VoteNet+TFA | 0.48 | 0.09 | 8.07 | 1.03 | 16.36 | 7.91 | 1.00 | 0.15 | 2.64 | 0.22 | 5.71 | 2.40 |
| VoteNet+TFA* | 10.38 | 3.96 | 23.77 | 9.83 | 26.02 | 13.96 | 5.12 | 0.95 | 16.23 | 3.72 | 21.89 | 6.76 |
| **Ours** | **15.34** | **8.25** | **31.25** | **16.01** | **32.25** | **19.52** | **11.01** | **2.21** | **21.14** | **8.39** | **28.52** | **12.35** |

Table 7: Results of VoteNet+TFA* on **FS-ScanNet** using mean Average Precision (mAP) at two different IoU thresholds of 0.25 and 0.50, denoted as $AP_{25}$ and $AP_{50}$.

**More Analysis on TFA.** TFA [39] is a simple yet effective method in few-shot 2D object detection. Specifically, TFA first trains a detector on the training set with abundant samples of base classes

| Method | 3-shot | | 5-shot | |
|---|---|---|---|---|
| | $AP_{25}$ | $AP_{50}$ | $AP_{25}$ | $AP_{50}$ |
| GroupFree + DeFRCN | 25.22 | 10.90 | 26.42 | 14.01 |
| GroupFree + FADI | 25.73 | 11.02 | 27.12 | 14.32 |
| 3DETR + DeFRCN | 26.01 | 10.95 | 26.88 | 14.45 |
| 3DETR + FADI | 26.24 | 11.12 | 26.93 | 15.22 |
| Ours | 31.25 | 16.01 | 32.25 | 19.52 |

Table 8: More Methods Borrowed From 2D Few-Shot Object detection

and then finetunes the classifier and the regressor on a small balanced set, which contains both base classes and novel classes. More details can be referred to in [39]. However, as shown in Table 1 and Table 2 in the main paper, VoteNet+TFA performs poorly in few-shot 3D object detection. As discussed in Section 4.3 in the main paper, the reason is that a few layers are finetuned by the samples of novel classes. A question arises here: why does TFA in few-shot 2D object detection only need to train the classifier and the regressor on the novel samples? We speculate that the large-scale pre-training on ImageNet [6] helps. To overcome this problem, in the first stage of TFA, we train the VoteNet on the training set with both novel classes and base classes. Then, in the second stage, we use a small balanced set containing both novel classes and base classes to finetune the classifier and the regressor. We denote this new baseline as VoteNet+TFA$^*$. As shown in Table 7, VoteNet+TFA$^*$ boosts the performance significantly. For example, in 3-shot in split-1 of FS-ScanNet, VoteNet+TFA$^*$ can achieve 26.02% $AP_{25}$ and 13.96% $AP_{50}$, while VoteNet+TFA only reaches 16.36% $AP_{25}$ and 7.91% $AP_{50}$.

**More Methods Borrowed From 2D Few-Shot Object detection.** We combine two SOTA 2D few-shot object detection techniques (i.e. DeFRCN [26], FADI [1]) and two SOTA 3D detectors (i.e. GroupFree [18], 3DETR [22]). These two few-shot techniques are plug-in-play modules and can be easily incorporated into the different detection architectures. We conducted this experiment on 3-shot and 5-shot in split-1 of FS-ScanNet. The results in Table 8 show that our method still surpasses these methods by a large margin. This is potentially because, in the 2D domain, they often build their model upon a large-scale pre-trained model on ImageNet. However, in the 3D community, there does not exist a large-scale dataset for model pre-training, which requires future investigations. Therefore, these 2D few-shot object detection techniques might not be directly transferable to the 3D domain. For future works, we might resort to the pre-training models in the 2D domain to facilitate the few-shot generalization on 3D few-shot learning and how these techniques can be combined with our method.

## 5    Concluding Remarks

In this paper, we have presented Prototypical VoteNet for FS3D along with a new benchmark for evaluation. Prototypical VoteNet enjoys the advantages of two new designs, namely Prototypical Vote Module (PVM) and Prototypical Head Module (PHM), for enhancing feature learning in the few-shot setting. Specifically, PVM exploits geometric prototypes learned from base categories to refine local features of novel categories. PHM is proposed to utilize class-specific prototypes to promote discriminativeness of object-level features. Extensive experiments on two new benchmark datasets demonstrate the superiority of our approach. We hope our studies on 3D propotypes and proposed new benchmark could inspire further investigations in few-shot 3D object detection.

## Acknowledgments and Disclosure of Funding

This work has been supported by Hong Kong Research Grant Council - Early Career Scheme (Grant No. 27209621), HKU Startup Fund, HKU Seed Fund for Basic Research, and Tencent Research Fund. Part of the described research work is conducted in the JC STEM Lab of Robotics for Soft Materials funded by The Hong Kong Jockey Club Charities Trust.

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
