# OpenReview forum: "Prototypical VoteNet for Few-Shot 3D Point Cloud Object Detection"
_NeurIPS.cc/2022/Conference — NeurIPS 2022 Accept_

### Official Review · Reviewer_ZU9c · 2022-07-08

**Rating:** 4
**Confidence:** 5
**Soundness:** 2 fair
**Presentation:** 3 good
**Contribution:** 2 fair

**Summary:**

This paper proposes a few-shot framework for 3D point cloud indoor object detection. They propose prototypical votenet to recognize
and localize novel instances based on the PVM and PHM modules. PVM leverages class-agnostic geometric prototypes learned from base classes to refine local features of novel categories. PHM is designed to utilize class prototypes to enhance the global feature of each object. This paper also provides two new benchmark datasets, FS-ScanNet and FS-SUNRGBD. They conduct extensive experiments to demonstrate the effectiveness of their method, which shows a promising performance compared to several self-designed baselines on two benchmark datasets.

**Questions:**

All the questions are mentioned in Weakness.

-I want to see more baseline approaches widely used in 2D few-shot object detection.

-I want to see the improvement of the whole class by the proposed model. (Compared with VoteNet).

If the authors solve my concerns, I will consider raising my score.

**Strengths And Weaknesses:**

Strengths:

-This paper is well-written and easy to follow.

-The idea is interesting. This paper considers utilizing geometric learning and object prototypes to assist few-shot 3D object detection, which will help our community.

-The results of their experiment are promising with a remarkable margin.

Weakness:

-The baselines of this paper are not well-designed. Few-shot object detection in 2D community is well-studied. Why didn't this paper compare some classical few-shot approaches in 2D community? I don't think it's difficult to transform some 2D techniques into the 3D domain in this task. It's difficult to judge their contribution based on these weak baselines.

-If you statistics them, both of these datasets i.e. ScanNet V2 and SUN RGB-D are unbalanced. In ScanNetV2, the number of chairs is 4357 and 113 of bathtubs. If this paper focuses on solving the imbalanced problem of different classes by considering the prototype learning, it would contribute more to the 3D community. So I want to see the improvement of fully training based on the votenet and proposed PVM&PHM. That means the base class is the whole class (18 classes) and no novel class.

---

> ### Author Response · Authors · 2022-08-02
> **Author Response to Reviewer ZU9c - Part 2**
>
> |    SUN RGB-D         | P (Original Dataset) | P (Original Dataset) |  10P  |  10P  |  25P  |  25P  |  50P  |  50P  |
> |-------------|:--------------------:|:--------------------:|:-----:|:-----:|:-----:|:-----:|:-----:|:-----:|
> | Method      |         AP25         |         AP50         |  AP25 |  AP50 |  AP25 |  AP50 |  AP25 |  AP50 |
> |   VoteNet   |         59.78        |         35.77        | 51.09 | 31.81 | 43.68 | 29.08 | 40.46 | 22.23 |
> |     Ours    |         60.34        |         36.80        | 51.85 | 32.98 | 44.66 | 31.93 | 41.84 | 25.04 |
> | Improvement |         0.56         |         1.03         |  0.96 |  1.17 |  0.98 |  2.85 |  1.38 |  2.81 |
>
> [5] Cui, Yin, Menglin Jia, Tsung-Yi Lin, Yang Song, and Serge Belongie. "Class-balanced loss based on effective number of samples." In Proceedings of the IEEE/CVF conference on computer vision and pattern recognition (CVPR), 2019.
>
> **Q3.Limitation Discussion.**
>
> **A3.** Thanks for your comments. Indeed, we have already included the limitation discussion in the supplementary material. We also copy it here for your reference.
>
> “Although the 3D cues of point clouds are more stable since they can get rid of some visual distractors, such as lighting and perspectives, some factors still impede the model from better generalization. For instance, in 3D scene understanding, if the point cloud in the training set is dense and that of the test set is sparse, a model often performs poorly, which can be treated as a cross-domain problem. Regarding few-shot 3D object detection, the performance might degrade if there is such a large domain gap between base classes and novel classes. Even though the basic geometric features are learned in the base classes, they might not be generalized well to the novel classes due to the difference in point cloud sparsity. The performance of this model has much room for improvement. One way to achieve better performance is large-scale pre-training. Large-scale pre-training enables the model to learn more generic features for transfer learning using limited samples, which benefits the community of 2D few-shot learning (i.e., ImageNet Pretraining). For future works, we might resort to the pre-training models in the 2D domain to facilitate the few-shot generalization on 3D few-shot learning and how these techniques can be combined with our method.“

---

> ### Author Response · Authors · 2022-08-02
> **Author Response to Reviewer ZU9c - Part 1**
>
> Thanks for your valuable comments and efforts in helping make our work better. We will explain your concerns and add them to our revised paper.
>
> **Q1.More baseline approaches widely used in 2D few-shot object detection.**
>
> **A1.** Thank you for the great suggestion. We will add the comparisons into our paper.
>
> We combine two SOTA 2D few-shot object detection techniques (i.e. DeFRCN [1], FADI [2]) and two SOTA 3D detectors (i.e. GroupFree [3], 3DETR [4]). These two few-shot techniques are plug-in-play modules and can be easily incorporated into the different detection architectures. We conducted this experiment on 3-shot and 5-shot in split-1 of FS-ScanNet. The results below show that our method still surpasses these methods by a large margin.
>
> This is potential because these 2D few-shot object detection techniques might not be directly transferable to the 3D domain. In the 2D domain, they often build their model upon a large-scale pre-trained model on ImageNet. However, in the 3D community, there does not exist a large-scale dataset for model pre-training,  which requires future investigations.
>
> |      | 3-shot |  3-shot     | 5-shot |   5-shot |
> |------------|:------:|:-----:|:------:|:-----:|
> | **Method**      |   **$AP_{25}$**  |  **$AP_{50}$** |  **$AP_{25}$**  | **$AP_{50}$**|
> |     VoteNet+ DeFRCN[1]    |  23.17 |  9.82 |  25.92 | 13.51 |
> |     VoteNet + FADI[2]     |  24.08 |  9.93 |  26.03 | 13.47 |
> | GroupFree[3] + DeFRCN [1] |  25.22 | 10.90 |  26.42 | 14.01 |
> |  GroupFree[3] +  FADI [2] |  25.73 | 11.02 |  27.12 | 14.32 |
> |   3DETR[4] + DeFRCN [1]   |  26.01 | 10.95 |  26.88 | 14.45 |
> |    3DETR[4] + FADI [2]    |  26.24 | 11.12 |  26.93 | 15.22 |
> |            Ours           |  31.25 | 16.01 |  32.25 | 19.52 |
>
> [1] Qiao, Limeng, Yuxuan Zhao, Zhiyuan Li, Xi Qiu, Jianan Wu, and Chi Zhang. "Defrcn: Decoupled faster r-cnn for few-shot object detection." In Proceedings of the IEEE/CVF International Conference on Computer Vision (ICCV), 2021.
>
> [2] Cao, Yuhang, Jiaqi Wang, Ying Jin, Tong Wu, Kai Chen, Ziwei Liu, and Dahua Lin. "Few-Shot Object Detection via Association and DIscrimination." Advances in Neural Information Processing Systems (NeurIPS), 2021.
>
> [3] Liu, Ze, Zheng Zhang, Yue Cao, Han Hu, and Xin Tong. "Group-free 3d object detection via transformers." In Proceedings of the IEEE/CVF International Conference on Computer Vision (ICCV), 2021.
>
> [4] Misra, Ishan, Rohit Girdhar, and Armand Joulin. "An end-to-end transformer model for 3d object detection." In Proceedings of the IEEE/CVF International Conference on Computer Vision (ICCV), 2021.
>
> **Q2. Improvement of the whole class by the proposed model. (Compared with VoteNet).**
>
> **A2.** To analyze the performance of the proposed model on the imbalance problem, we conduct experiments using all the classes. Note that we conduct the experiments not only on the original ScanNet V2 and SUN RGB-D datasets, but also on their more unbalanced counterparts.
>
> We follow the benchmark [5] to create these counterparts: 1) sorting the classes in descending order according to number of samples in each class, then we have $n_i > n_j$ if $i < j$, where $n$ is the number of samples, $i$ and $j$ denote the index of the classes. 2) reducing the number of training samples per class according to an exponential function $n=n_i*u^i$, where $u \in (0,1)$. The test set remains unchanged.
>
> According to the benchmark [5], we define the imbalance factor of a dataset as the number of training samples in the largest class divided by the smallest. Note that we use P as the value of the imbalance factor in the original ScanNet V2 and SUN RGB-D datasets. Additionally, we add another three sets, whose values of imbalance factor are 10P, 25P and 50P, for both ScanNet V2 and SUN RGB-D datasets.
>
> As shown in the table below, the experimental results indicate that our proposed method consistently outperforms the baseline VoteNet by a large margin, especially when the dataset is severely unbalanced. This is because the proposed method develops a more generic vote module by learning geometric prototypes, and leverages class-specific prototypes to enhance the discriminative feature learning. Note that on the original dataset (P), our model also outperforms the baseline.
>
> |     ScanNet V2        | P (Original Dataset) | P (Original Dataset) |  10P  |  10P  |  25P  |  25P  |  50P  |  50P  |
> |-------------|:--------------------:|:--------------------:|:-----:|:-----:|:-----:|:-----:|:-----:|:-----:|
> | Method      |         AP25         |         AP50         |  AP25 |  AP50 |  AP25 |  AP50 |  AP25 |  AP50 |
> |   VoteNet   |         62.34	        |         40.82        | 52.06 | 35.64 | 43.12 | 27.13 | 40.01 | 26.77 |
> |     Ours    |         62.59        |         41.25        | 52.60 | 36.87 | 44.53 | 29.17 | 41.99 | 29.01 |
> | Improvement |         0.25         |         0.43         |  0.54 |  1.23 |  1.41 |  2.04 |  1.98 |  2.24 |

---

> > ### Comment · Reviewer_ZU9c · 2022-08-05
> > **Questions of Rebuttal by Reviewer ZU9c**
> >
> > Thanks for the high-quality rebuttal. If all the experiments were reliable, this would be a heavy rebuttal. However,  I am confused about the performance of group-free and 3DETR. In regular 3D object detection, group-free is better than 3DETR with the same backbone (PointNet++). Why is 3DETR better than Group-Free in your experiments (*e.g.*, few-shot setting)? Besides that, i am also confused about the results of Q2. Why is the performance gap the same in map@0.25 and map@0.50.
> > |ScanNet V2	|P (Original Dataset)	|P (Original Dataset)	|10P	|10P	|25P	|25P	|50P	|50P|
> > |:---------------:|:--------------------------:|:---------------------------:|:----:|:----:|:----:|:----:|:----:|:----:|
> > Method	        |AP$_{25}$                              |AP$_{50}$              |AP$_{25}$|AP$_{50}$|AP$_{25}$|AP$_{50}$|AP$_{25}$|AP$_{50}$|
> > VoteNet	| 62.34	|40.82	|53.82	|34.93	|45.43	|28.01	|39.22	|24.82|
> > Ours	|64.37	|42.96	|57.05	|38.01	|50.22	|33.27	|45.43	|31.08|
> > Improvment        |2.03         |2.14        |3.23        |3.08         |4.79        |5.26        |6.21         |6.26|

---

> > > ### Author Response · Authors · 2022-08-07
> > > **Author Response to Reviewer ZU9c - Part 3**
> > >
> > > Thank you so much for your time and efforts on assessing our paper. Your valuable comments help improve our paper a lot.
> > >
> > > **Q4. Group-Free vs 3DETR in 3D few-shot detection.**
> > >
> > > **A4.** Thanks for your comment. As shown in the following Table (3-shot in split-1 of FS-ScanNet), although Group-Free performs slightly worse than 3DETR on the novel categories, we found that on the base categories (where abundant training samples exist), the Group-Free based method is better than the 3DETR based method. This echoes the performance of the original papers in the fully-supervised setting.
> > >
> > > One possible reason for the lower performance of Group-Free in the few-shot setting is that the learnable proposal candidate generation stage might be biased toward base categories, as elaborated below.
> > >
> > > Specifically, Group-Free first obtains initial object candidates using k-Closest Points Sampling (default implementation) which needs a learned objectness classifier to predict the probability of each point belonging to a ground-truth object candidate. Then, points with  high classification scores are further used as queries for the second stage (i.e., decoding process) to predict 3D boxes. Note that in the few-shot setting, the learning of objectness classifiers can be easily dominated by the base classes due to the small number of novel objects. Therefore, the proposal candidate stage might be easily biased toward base classes, which impedes the decoding of novel samples.
> > >
> > > As for 3DETR, a randomly sampling method is used to generate initial object candidates. Therefore, it is not biased towards base classes, which can lead to better generalization ability on the few-shot problem.  In the future work, we will equip Group-Free with a randomly sampling method to see whether it benefits Group-Free on the few-shot problem or not.
> > >
> > > |                           |  Novel | Novel |  Base |  Base |
> > > |---------------------------|:------:|:-----:|:-----:|:-----:|
> > > | Method                    |  AP25  |  AP50 |  AP25 |  AP50 |
> > > |  GroupFree[3] +  FADI [2] |  25.73 | 11.02 | 64.86 | 44.01 |
> > > |    3DETR[4] + FADI [2]    |  26.24 | 11.12 | 62.56 | 42.10 |
> > > | GroupFree[3] + DeFRCN [1] |  25.22 | 10.90 | 64.95 | 44.28 |
> > > |   3DETR[4] + DeFRCN [1]   |  26.01 | 10.95 | 62.43 | 42.26 |
> > >
> > > **Q5. The performance of the imbalance problem.**
> > >
> > > **A5.**  We are grateful for your careful review. Much appreciate your comments. After receiving your comment, we carefully checked all codes and found one error in our evaluation code for the imbalance problem. The updated results are shown in the following Table. Note that we achieve comparable performance in the original dataset setting. With the imbalance becoming more severe (e.g., 25P, 50P), our approach outperforms the baseline more.
> > >
> > > Note that our focus is on few-shot 3D object detection, where representation learning of new categories becomes the top consideration of algorithm design. This few-shot problem is more useful for scenarios where many new categories appear frequently and require the system to quickly adapt to recognize them.
> > >
> > > However, the long-tailed problem focuses on how to learn good representations and classifiers that can deliver good performance for both head and tail categories. We believe that dedicated designs can further improve the performance of long-tailed 3D object detection. We will also add the results and analysis for the long-tailed setting in our paper and hope to inspire more future investigations.
> > >
> > > The new testing logs can be seen at the anonymous link:
> > > https://drive.google.com/drive/folders/18S2SxEEtqYGb1Qb3njylWDqGDG2wv8Mo?usp=sharing
> > >
> > > |     ScanNet V2        | P (Original Dataset) | P (Original Dataset) |  10P  |  10P  |  25P  |  25P  |  50P  |  50P  |
> > > |-------------|:--------------------:|:--------------------:|:-----:|:-----:|:-----:|:-----:|:-----:|:-----:|
> > > | Method      |         AP25         |         AP50         |  AP25 |  AP50 |  AP25 |  AP50 |  AP25 |  AP50 |
> > > |   VoteNet   |         62.34	        |         40.82        | 52.06 | 35.64 | 43.12 | 27.13 | 40.01 | 26.77 |
> > > |     Ours    |         62.59        |         41.25        | 52.60 | 36.87 | 44.53 | 29.17 | 41.99 | 29.01 |
> > > | Improvement |         0.25         |         0.43         |  0.54 |  1.23 |  1.41 |  2.04 |  1.98 |  2.24 |
> > >
> > >
> > > |    SUN RGB-D         | P (Original Dataset) | P (Original Dataset) |  10P  |  10P  |  25P  |  25P  |  50P  |  50P  |
> > > |-------------|:--------------------:|:--------------------:|:-----:|:-----:|:-----:|:-----:|:-----:|:-----:|
> > > | Method      |         AP25         |         AP50         |  AP25 |  AP50 |  AP25 |  AP50 |  AP25 |  AP50 |
> > > |   VoteNet   |         59.78        |         35.77        | 51.09 | 31.81 | 43.68 | 29.08 | 40.46 | 22.23 |
> > > |     Ours    |         60.34        |         36.80        | 51.85 | 32.98 | 44.66 | 31.93 | 41.84 | 25.04 |
> > > | Improvement |         0.56         |         1.03         |  0.96 |  1.17 |  0.98 |  2.85 |  1.38 |  2.81 |

---

> > > > ### Comment · Reviewer_ZU9c · 2022-08-08
> > > > **Response of Rebuttal by Reviewer ZU9c**
> > > >
> > > > Thanks for your careful reply. You're very responsible and conscientious. Even though the responses didn't completely solve my concerns, such as limited improvement compared to Votenet on the full dataset, I would still raise my score (4->5) for your effort and honesty. By the way, I hope you will pay attention to the normal or long-tail settings rather than few-shot task. Please don't mind, in my opinion, few-shot is of little significance and limited application. Thanks for your careful reply again. Hope our community can have more responsible reviews and responses. Good Luck!
> > > >
> > > > Please update the new results in the final version. Thanks.

---

> > > > > ### Author Response · Authors · 2022-08-08
> > > > > **Author Response**
> > > > >
> > > > > Dear Reviewer ZU9c,
> > > > >
> > > > > We sincerely thank the reviewer for the constructive feedback and support. We will update the new results in our final version. Thanks again for your time and efforts in assessing our paper.

---

> > > > > ### Comment · Reviewer_ZU9c · 2022-08-08
> > > > > **Response of Rebuttal by Reviewer ZU9c**
> > > > >
> > > > > After re-reading your paper and carefully thinking about it, I still don't understand why your method is not better than Votenet. There should be consistency between the performance of few-shot and unbalance setting. As far as I know, both datasets are very unbalanced, (*e.g.*, the number of chairs is 4357 and 113 of bathtubs. ). That phenomenon makes me confused. Considering the previous bug in the evaluation code, it's very hard for me to judge the reliability of this paper. Terribly sorry, I may consider revoking my previous decision of raising the score.

---

> > > > > > ### Author Response · Authors · 2022-08-08
> > > > > > **Author Response to Reviewer ZU9c - Part 4**
> > > > > >
> > > > > > **Q6. The consistency between the performance of few-shot and unbalance setting.**
> > > > > >
> > > > > > **A6.** Thanks for your time and comments. You are very responsible and we appreciate your efforts a lot.
> > > > > >
> > > > > > Here we would like to further clarify the difference between the few-shot learning and the imbalance problem in ScanNet and SUN RGB-D, and address the consistency concerns of performance improvement.
> > > > > >
> > > > > > 1. **Few-shot learning focuses on feature learning under the circumstance of severely scarce data**. For example, in the widely-used 2D few-shot learning benchmark [1], the instance number for novel classes is from 1 to 5, as studied in our paper. When data is extremely scarce, the feature learning process will suffer from serious overfitting. As shown in the following table, when the bathtub class has only a few samples from 1  to 5, the original VoteNet can not learn well, while our proposed method can outperform the baseline VoteNet by a large margin.
> > > > > > However, in the original dataset, which is an imbalanced dataset, with the baseline VoteNet, the performance of the bathtub category is already pretty high (as shown in the table below, and you can check it by this link [2] in epoch 21), approaching perfect results. Therefore, our model can only deliver a smaller improvement.
> > > > > > |  Bathtub performance        | VoteNet  | VoteNet |  Ours |  Ours |
> > > > > > |-----------------------------|:--------:|:-------:|:-----:|:-----:|
> > > > > > |                             |   AP25   |   AP50  |  AP25 |  AP50 |
> > > > > > |            1-shot           |   0.74   |   0.01  |  9.01 |  7.63 |
> > > > > > |            3-shot           |   12.96  |   1.26  | 22.96 |  8.60 |
> > > > > > |            5-shot           |   17.57  |   3.25  | 30.33 | 12.87 |
> > > > > > | **113-shot (original dataset)** |   **91.86**  |  **84.48**  | **92.57** | **85.36** |
> > > > > >
> > > > > > 2. The **imbalance problem** focuses on how to learn good representations and classifiers that can deliver good performance for both head and tail categories. **It is designed to address the problem of unequal sample sizes within the dataset, but not necessarily the few-shot problem.** For example, in ScanNet V2,  the category (Bathtub) with the minimum samples has 113 instances, and the performance for this class already achieves a very high performance, as shown in the table above.
> > > > > > **To our best knowledge, all few-shot benchmarks do not use such a large number of samples for novel classes, so this is no longer a problem of few-shot learning.** While our method is specifically designed for few-shot 3D detection, our model may not improve performance on the original dataset if it already contains a sufficient number of training instances (e.g., ScanNet) for all classes.
> > > > > >
> > > > > > 3. In a widely-used long-tailed learning benchmark [3], it sets the many-shot classes (classes each with over training 100 samples), medium-shot classes (classes each with 20∼100 training samples) and few-shot classes (classes under 20 training samples). Therefore, this is another evidence that we cannot consider a dataset with a minimum sample number greater than 100 as a few-shot problem.
> > > > > >
> > > > > > 4. In terms of **performance consistency**, as shown in the table in Q5, with the imbalance becoming more severe (e.g., 25P, 50P), our approach outperforms the baseline more.  In the even more extreme case, the imbalance problem will degrade to few-shot learning, and our proposed method will benefit more.
> > > > > >
> > > > > > 5. We would also share our understanding on **the practical significance of few-shot learning**. For some scenarios, such as autonomous driving and the medical domain, it’s very challenging to gather many samples (such as car accidents or rare diseases) where few-shot learning will help a lot.
> > > > > > On the other hand, in the open-world setting, considering the number of potential classes, few-shot learning will also help a lot to significantly alleviate the burden of data collection and annotation. Besides, an existing study [3] already demonstrated that the open-world long-tailed recognition will encounter the few-shot problem and develop methods inspired by the few-shot domain to address the issue.
> > > > > >
> > > > > > Thanks again for your time and efforts in assessing our paper. **Our code, as well as the new benchmark, will be released to facilitate future works.**
> > > > > >
> > > > > > [1] Sung, Flood, Yongxin Yang, Li Zhang, Tao Xiang, Philip HS Torr, and Timothy M. Hospedales. "Learning to compare: Relation network for few-shot learning." In Proceedings of the IEEE conference on computer vision and pattern recognition, pp. 1199-1208. 2018.
> > > > > >
> > > > > > [2] https://download.openmmlab.com/mmdetection3d/v1.0.0_models/votenet/votenet_8x8_scannet-3d-18class/votenet_8x8_scannet-3d-18class_20210823_234503.log.json, Log of the publicly trained VoteNet by OpenMMab.
> > > > > >
> > > > > > [3] Liu, Ziwei, Zhongqi Miao, Xiaohang Zhan, Jiayun Wang, Boqing Gong, and Stella X. Yu. "Large-scale long-tailed recognition in an open world." In Proceedings of the IEEE/CVF Conference on Computer Vision and Pattern Recognition, pp. 2537-2546. 2019.

---

### Official Review · Reviewer_Eu2W · 2022-07-11

**Rating:** 7
**Confidence:** 4
**Soundness:** 4 excellent
**Presentation:** 3 good
**Contribution:** 3 good

**Summary:**

This paper proposes a new task, few-shot 3d point cloud object detection. It is a combination of two well-studied topics, few-shot learning and 3d point cloud detection. And naturally, two well-known methods from both sides, prototypical learning and VoteNet, are combined to tackle the new task.

While the combination seems straight forward, the paper further shows that it is not trivial. The implementation of PVM and PHM disentangles the feature embedding and detection. And successfully

**Questions:**

Questions are listed in weaknesses.

**Limitations:**

The limitations of the work has not been addressed.

**Strengths And Weaknesses:**

It is overall a good work. Strengths are already covered in summary. Here are some weaknesses.

1. As a general problem when a few-shot detection benchmark is composed, do we have novel classes in training set but annotated as background? What is is ratio?

2. Can you visualize some typical structures of geometric prototypes? For example, the neighborhood of the point with features close to each prototype?

3. The abbreviation FSOD is usually used for few-shot object detection (2D). As a novel task for 3D, the name could be more specific, such as FS3D.

---

> ### Author Response · Authors · 2022-08-02
> **Author Response to Reviewer Eu2W**
>
> Thanks for your valuable comments and efforts in helping make our work better. We will explain your concerns and add them to our revised paper.
>
> **Q1. As a general problem when a few-shot detection benchmark is composed, do we have novel classes in the training set but annotated as background? What is the ratio?**
>
> **A1.** Because of the separation between instance and background in the 3D point cloud, the removal of the novel samples does not affect the global scene. To ensure that there are only a few (k) instances for novel categories, we artificially remove those samples. Therefore, considering the large sample size of base classes, this ratio is that is k-shot $\times$ number of novel categories $/$ total number of samples including base and novel, which is nearly zero.
>
> **Q2. Can you visualize some typical structures of geometric prototypes? For example, the neighborhood of the point with features close to each prototype?**
>
> **A2.** The anonymous link for visualization:
> https://drive.google.com/file/d/1vu4qMcsmYlau-518PYSbqbm-6OYnrBVM/view
>
> Thank you for this insightful suggestion. Here, we visualize the relation between the learned geometric prototype and the 3D points by searching points with features that are similar to a given geometric prototype. First, we feed object point clouds to a trained Prototypical VoteNet. Second, for each point feature, we can search for its most similar prototype. If the similarity is above a threshold, we can assign the point to that prototype. Third, we use a density-based clustering algorithm to cluster the point groups, and we draw the minimum 3D bounding box around each point group.
>
> As shown in the figure, all the red bounding boxes within each subfigure belong to the same prototype. The result shows that in each subfigure, the enclosed geometric structures are similar. For example, subfigure (a) illustrates that the prototype learns the feature of corners, while subfigure (b) shows that the prototype learns the long stick.
>
> **Q3. The abbreviation FSOD is usually used for few-shot object detection (2D). As a novel task for 3D, the name could be more specific, such as FS3D.**
>
> **A3.** Thanks for the valuable suggestion. We have changed the abbreviation to FS3D in our rebuttal revision.
>
> **Q4.  Limitation Discussion.**
>
> **A4.** Due to limited space, we have included limitation discussion in the supplementary material Section A.7. We are sorry for not stating it in the manuscript. We copy the limitation analysis here for your reference.
>
> “Although the 3D cues of point clouds are more stable since they can get rid of some visual distractors, such as lighting and perspectives, some factors still impede the model from better generalization. For instance, in 3D scene understanding, if the point cloud in the training set is dense and that of the test set is sparse, a model often performs poorly, which can be treated as a cross-domain problem. Regarding few-shot 3D object detection, the performance might degrade if there is such a large domain gap between base classes and novel classes. Even though the basic geometric features are learned in the base classes, they might not be generalized well to the novel classes due to the difference in point cloud sparsity. The performance of this model has much room for improvement. One way to achieve better performance is large-scale pre-training. Large-scale pre-training enables the model to learn more generic features for transfer learning using limited samples, which benefits the community of 2D few-shot learning (i.e., ImageNet Pretraining). For future works, we might resort to the pre-training models in the 2D domain to facilitate the few-shot generalization on 3D few-shot learning and how these techniques can be combined with our method. “

---

### Official Review · Reviewer_ymyp · 2022-07-11

**Rating:** 4
**Confidence:** 4
**Soundness:** 3 good
**Presentation:** 3 good
**Contribution:** 2 fair

**Summary:**

The paper presents a method for few-shot 3D point cloud object detection. The method extends VoteNet by incorporating two new modules, prototypical vote module (PVM) and prototypical head module (PHM). The experimental results show that the proposed prototypical VoteNet improves the performance for few-shot t 3D point cloud object detection.

**Questions:**

- How are the new components different from existing work, such as [37]?
- How are the hyperparameters such as alpha_1, alpha_2, etc. set?


**Limitations:**

The limitations of the paper are not sufficiently discussed. Given the relatively low performance, the paper should include more discussions of limitations beyond just resolution difference.

**Strengths And Weaknesses:**

Strengths:

- The paper addresses a new problem, which is potentially useful.
- The method is plausible and seems to work better than existing methods.

Weaknesses:

- The technical components are largely adapted from existing papers, so technical novelty is somewhat limited.
- The few-shot setting is not sufficiently convincing. The categories chosen for few-shot learning are not really rare categories, so it is not sufficiently meaningful. The performance is also rather low for practical use.
- The datasets are straightforward adaptations of well-known datasets, so they should be declared as a contribution.
- The paper only compares the baseline VoteNet, not follow-up improvements which could give better performance.

---

> ### Author Response · Authors · 2022-08-02
> **Author Response to Reviewer ymyp - Part 2**
>
> **Q3. The datasets are straightforward adaptations of well-known datasets, so they should be declared as a contribution.**
>
> **A3.** Thanks for the comments. We would like to emphasize that our contribution is not to collect and annotate datasets, but to standardize a few-shot dataset setting and set up a benchmark where several baseline methods have been implemented.  This benchmark can become the basis for future investigations and inspire follow-up works.
>
> **Q4. The paper only compares the baseline VoteNet, not follow-up improvements which could give better performance.**
>
> **A4.** Thank you for the comments. We conduct experiments on more advanced object detectors (i.e., GroupFree [3], 3DETR [4]) and SOTA 2D few-shot detection techniques (i.e., DeFRCN [5], FADI [6]). We conducted this experiment on 3-shot and 5-shot in split-1 of FS-ScanNet. The experimental results are shown below. Our method still surpasses these methods with a large margin.
>
> This is potential because these 2D few-shot object detection techniques might not be directly transferable to the 3D domain. In the 2D domain, they often build their model upon a large-scale pre-trained model on ImageNet. However, in the 3D community, there does not exist a large-scale dataset for model pre-training,  which requires future investigations.
>
> Moreover, comparing the performance of different backbone detectors, we observe that a better detection architecture does not bring large performance gains in the few-shot 3D detection scenario.  The most challenging issue for few-shot 3D object detection still lies in how to learn effective representation if only a few training samples are provided. The architecture don’t help much if the model cannot effectively extract features to represent novel categories with only a few samples.
>
>
> |      | 3-shot |  3-shot     | 5-shot |   5-shot |
> |------------|:------:|:-----:|:------:|:-----:|
> | **Method**      |   **$AP_{25}$**  |  **$AP_{50}$** |  **$AP_{25}$**  | **$AP_{50}$**|
> |     VoteNet+ DeFRCN[5]    |  23.17 |  9.82 |  25.92 | 13.51 |
> |     VoteNet + FADI[6]     |  24.08 |  9.93 |  26.03 | 13.47 |
> | GroupFree[3] + DeFRCN [5] |  25.22 | 10.90 |  26.42 | 14.01 |
> |  GroupFree[3] +  FADI [6] |  25.73 | 11.02 |  27.12 | 14.32 |
> |   3DETR[4] + DeFRCN [5]   |  26.01 | 10.95 |  26.88 | 14.45 |
> |    3DETR[4] + FADI [6]    |  26.24 | 11.12 |  26.93 | 15.22 |
> |            Ours           |  31.25 | 16.01 |  32.25 | 19.52 |
>
> [3] Liu, Ze, Zheng Zhang, Yue Cao, Han Hu, and Xin Tong. "Group-free 3d object detection via transformers." In Proceedings of the IEEE/CVF International Conference on Computer Vision (ICCV), 2021.
>
> [4] Misra, Ishan, Rohit Girdhar, and Armand Joulin. "An end-to-end transformer model for 3d object detection." In Proceedings of the IEEE/CVF International Conference on Computer Vision (ICCV), 2021.
>
> [5] Qiao, Limeng, Yuxuan Zhao, Zhiyuan Li, Xi Qiu, Jianan Wu, and Chi Zhang. "Defrcn: Decoupled faster r-cnn for few-shot object detection." In Proceedings of the IEEE/CVF International Conference on Computer Vision (ICCV), 2021.
>
> [6] Cao, Yuhang, Jiaqi Wang, Ying Jin, Tong Wu, Kai Chen, Ziwei Liu, and Dahua Lin. "Few-Shot Object Detection via Association and DIscrimination." Advances in Neural Information Processing Systems (NeurIPS), 2021.
>
> **Q6. How are the hyperparameters such as alpha_1, alpha_2, etc. set?**
>
> **A6.** We follow the implementation of the released code [7] and don't make any adjustments to these hyperparameters in all our experiments.
>
> [7] Qi, Charles R., Or Litany, Kaiming He, and Leonidas J. Guibas. "Deep hough voting for 3d object detection in point clouds." In Proceedings of the IEEE/CVF International Conference on Computer Vision (ICCV), 2019.
>
> **Q7. Limitation Discussion**
>
> **A7.** Thanks for pointing this out. Beyond resolution difference, the relatively low performance is another limitation. Besides only a few training samples available, another reason for this problem is the lack of large-scale pre-training in the 3D domain. Large-scale pre-training enables the model to learn more generic features for transfer learning using limited samples, which benefits the community of 2D few-shot learning (i.e., ImageNet Pretraining). For future works, we might resort to the pre-training models in the 2D domain to facilitate the few-shot generalization on 3D few-shot learning, and how these techniques can be combined with our method.
>
> We will add this discussion in our paper.

---

> ### Author Response · Authors · 2022-08-02
> **Author Response to Reviewer ymyp - Part 1**
>
> Thanks for your valuable comments and efforts in helping make our work better. We will explain your concerns and add them to our revised paper.
>
> **Q1. The technical components are largely adapted from existing papers, so technical novelty is somewhat limited. How are the new components different from existing work, such as [37]?**
>
> **A1.** Our technical contribution majorly lies in how to make 3D object detection work when only a few training samples are available for a novel class. This is the first investigation in this area, and a challenging problem as few samples are not sufficient for learning useful feature representations. To this end, we propose two modules to enhance feature representation learning from a local and global perspective: 1) based on our motivation that 3D primitives to constitute objects can be shared among different categories, PVM is developed to learn robust class-agnostic geometric prototypes from base categories with abundant training samples, which are further transferred to enhance local feature learning of novel categories; 2) to improve discriminativeness of class categorization, we design class-specific prototypes which can be treated as a template for classifying novel categories and are used to refine the global features of samples.
>
> Our work is different from existing work [37] (Attention is All you Need) in the following folds: 1)  [37] focuses on a new method for NLP tasks and proposes to use attention as the core representation learning block for the whole network, i.e. stacking many attention layers.  The network is trained in a data-rich setting. Later on, multi-head self-attention becomes a generic block just like the residual block in resnet.  2) In contrast, our focus is on few-shot 3D object detection, that is how to enhance representation learning when only a few samples are available. Our core insight is to develop class-agnostic geometric prototypes and class-specific prototypes to enhance local and global feature representation learning, respectively. To make prototypes interact with feature representations, we leverage the multi-head attention block which computes affinity and aggregate prototypes to refine local and global features.
>
> Note that our highlight is not on the design of the multi-head attention module (which is the contribution of [37]) but how we develop prototypes and employ them to improve feature representation learning in the few-shot setting.
>
> [37] Vaswani, Ashish, Noam Shazeer, Niki Parmar, Jakob Uszkoreit, Llion Jones, Aidan N. Gomez, Łukasz Kaiser, and Illia Polosukhin. "Attention is all you need." Advances in neural information processing systems (NeurIPS), 2017.
>
> **Q2. The categories chosen for few-shot learning are not really rare categories, so it is not sufficiently meaningful. The performance is also rather low for practical use.**
>
> **A2.** This paper aims to investigate the problem of 3D object detection with a small number of samples for novel categories. Therefore, we randomly set some base/novel splits in ScanNet and SUNRGBD, which are the well-known datasets in 3D object detection. This is also the widely-adopted splitting method in the few-shot community.
>
> Performance:
> * Please note that although few-shot object detection in the 2D community is relatively well-studied, the performance of 2D few-shot object detection [1, 2] has a large gap ( i.e., around 30% mAP) compared to the full-supervision counterparts.
> * As an exploratory work, we are the first attempt to study Few-Shot 3D Point Cloud Object Detection and set up the basic benchmark for future studies. We believe that there are more chances in 3D few-shot learning since it suffers less influence on distortion, scale ambiguity, and texture variations. We hope to inspire more future studies.
> * The few shot learning setting that we study also has high practical impacts. The recent success of 3D detectors relies heavily on a huge amount of training data with accurate bounding box annotations. However, in many practical applications such as self-driving vehicles and robot manipulations, recognition systems need to rapidly adapt and recognize some never-before-seen objects from a very limited number of examples. We believe few-shot 3D recognition is one important step toward recognition in the open world as there are so many categories in our 3D world that we cannot afford to annotate them all with abundant samples.
>
> [1] Wang, Xin, Thomas E. Huang, Trevor Darrell, Joseph E. Gonzalez, and Fisher Yu. "Frustratingly simple few-shot object detection." In Proceedings of the International Conference on Machine Learning (ICML), 2020.
>
> [2] Qiao, Limeng, Yuxuan Zhao, Zhiyuan Li, Xi Qiu, Jianan Wu, and Chi Zhang. "Defrcn: Decoupled faster r-cnn for few-shot object detection." In Proceedings of the IEEE/CVF International Conference on Computer Vision (ICCV), 2021.

---

> ### Author Response · Authors · 2022-08-09
> **Author Response to Reviewer ymyp**
>
> Dear Reviewer ymyp,
>
> Thank you so much for your time and efforts in assessing our paper. Hope our rebuttal has addressed your concerns. We are happy to discuss with you further if you still have other concerns. Thanks for helping improve our paper.

---

### Official Review · Reviewer_M2Gz · 2022-07-12

**Rating:** 7
**Confidence:** 2
**Soundness:** 2 fair
**Presentation:** 4 excellent
**Contribution:** 3 good

**Summary:**

This work explores the problem of few-shot learning in 3D object detection. It appears to be the first work in the field, and contributes 2 dataset-settings, based on SUNRGB and ScanNet, as well as 4 benchmarks, mostly based on VoteNet. The authors contribute their own method, which extends VoteNet to incorporate prototypes in each of the feature representation “stages”. The prototypes are moving averages of the closest features. The method is simple, yet effective, showing strong performance gains compared to the other VoteNet baselines.

**Questions:**

Were other methods of updating the prototypes tried? Does setting the prototype at the end (no updates) perform well? (not necessarily a weakness, just a curious possible experiment)

**Limitations:**

The authors did not discuss any limitations.

**Strengths And Weaknesses:**

This work, to the best of my knowledge, is one of the first works to tackle the problem of few-shot learning in a 3D object detection setting. In addition to their own method, the authors also contribute a set of benchmarks and a proposed dataset setting to evaluate future few-shot learning methods. This is a good contribution to the community as a whole. The method proposed by the author appears to achieve strong performance compared to proposed benchmarks. The method appears to be informed, and empirical evidence backs up each module’s necessity. By initializing the prototypes used for the features (“geometric”) and for the classes with the feature-space centroids and updating with a moving average, it seems to prevent the few-shot learner from overfitting as much (table 1,2). Experimentally, the ablation section is detailed and I do not have any questions after reading it. Nice to see the prototypes didn’t collapse (fig 3).

The conclusion in section 3.2, L162 that the prototypes are learning basic 3D geometric shapes is a bit misleading, since distribution of features in any pre-trained detection model should be similar for similar objects as well (fig 3). Considering the prototypes are updated in the feature space, there is nothing to indicate that there is 3D geometric interpretation; one way to visualize if it truly is learning basic geometric shapes is to optimize a shape input to maximize one prototype. This would give a better “geometric” interpretation of what that prototype is representing. However, this is not necessarily a weakness, since feature-space prototypes are just as useful, but I would recommend either changing the claim, or backing up the claim with visualized representations of what each prototype corresponds to in 3D input space.
One weakness is the benchmarks proposed are all built on top of VoteNet. A benchmark that would be good to include is KNN assignment to the closest support class. This would be able to leverage other detectors.

Miscellaneous:
L121: “Taking a point cloud scene P_i as input… localize and categorize…” → “VoteNet takes a point cloud scene P_i as input and localizes and categorizes…”
Fig 4 caption: “Tsne” → “t-SNE”
L225: “Till now” → “Until now”

---

> ### Author Response · Authors · 2022-08-02
> **Author Response to Reviewer M2Gz - Part 2**
>
> **Q5. The authors did not discuss any limitations.**
>
> **A5.**  Due to limited space, we have included the limitation discussion in the supplementary material. We are sorry for not stating it in the manuscript. We copy the limitation discussion in Section A.7 here for your reference.
>
> “Although the 3D cues of point clouds are more stable since they can get rid of some visual distractors, such as lighting and perspectives, some factors still impede the model from better generalization. For instance, in 3D scene understanding, if the point cloud in the training set is dense and that of the test set is sparse, a model often performs poorly, which can be treated as a cross-domain problem. Regarding few-shot 3D object detection, the performance might degrade if there is such a large domain gap between base classes and novel classes. Even though the basic geometric features are learned in the base classes, they might not be generalized well to the novel classes due to the difference in point cloud sparsity. The performance of this model has much room for improvement. One way to achieve better performance is large-scale pre-training. Large-scale pre-training enables the model to learn more generic features for transfer learning using limited samples, which benefits the community of 2D few-shot learning (i.e., ImageNet Pre-training). For future works, we might resort to the pre-training models in the 2D domain to facilitate the few-shot generalization on 3D few-shot learning and how these techniques can be combined with our method.“
>
> **Q6. Miscellaneous.**
>
> **A6.** Thanks for pointing these out. We have corrected these problems in our rebuttal revision.

---

> > ### Comment · Reviewer_M2Gz · 2022-08-09
> > **Post-rebuttal**
> >
> > Thank you for the detailed and through response to my concerns. I have no additional questions, and the results from the extra baselines are quite convincing. Thank you for putting this together in this limited time. I look forward to the final version of the work.

---

> ### Author Response · Authors · 2022-08-02
> **Author Response to Reviewer M2Gz - Part 1**
>
> Thanks for your valuable comments and efforts in helping make our work better. We will explain your concerns and add them to our revised paper.
>
> **Q1. Visualize whether it is learning basic geometric shapes.**
>
> **A1.** The anonymous link for visualization:
> https://drive.google.com/file/d/1vu4qMcsmYlau-518PYSbqbm-6OYnrBVM/view.
>
> Thank you for this insightful suggestion. Here, we visualize the relation between the learned geometric prototypes and the 3D points by searching points with features that are similar to a given geometric prototype. First, we feed object point clouds to a trained Prototypical VoteNet. Second, for each point feature, we can search for its most similar prototype. If the similarity is above a threshold, we can assign the point to that prototype. Third, we use a density-based clustering algorithm DBSCAN to cluster the point groups, and we draw the minimum 3D bounding box around each point group.
>
> As shown in the figure, all the red bounding boxes within each subfigure belong to the same prototype. The result shows that in each subfigure, the enclosed geometric structures are similar. For example, subfigure (a) illustrates that the prototype learns the feature of corners, while subfigure (b) shows that the prototype learns the long stick.
>
> **Q2. KNN assignment and other detectors.**
>
> **A2.** Thank you for this great suggestion.
> * We apply KNN assignment to VoteNet and two SOTA 3D detectors GroupFree [1] and 3DETR [2]. We conducted this experiment on 3-shot and 5-shot in split-1 of FS-ScanNet. The KNN assignment is realized by calculating the distance between each object feature and features of all training objects in the classification step, and assigning the sample to the class based on voting from its k-nearest objects of the training set. Here, we take k as one since we find increasing the value k doesn’t improve performance.  The results are shown in the following Table.
> * Comparing the performance of “Baseline VoteNet”, “VoteNet + KNN” and “ours”, we see that the non-parametric KNN classifier will not help improve few-shot learning much (“VoteNet” vs “VoteNet+KNN”).
> * Comparing the performance of different backbone detectors (“VoetNet+KNN”, “GroupFree + KNN”, and “3DETR+KNN”), we observe that a better detection architecture does not bring large performance gains in the few-shot 3D detection scenario.
> * The most challenging issue for few-shot 3D object detection still lies in how to learn effective representation if only a few training samples are provided. The classifier and architecture don’t help much if the model cannot effectively extract features to represent novel categories with only a few samples.
> * We will add the comparison and analysis in the paper.
>
>
> |                 | **3-shot** | **3-shot**    | **5-shot** |  **5-shot**|
> |:---------------:|:------:|:-----:|:------:|:-----:|
> |   **Method**      |  **$AP_{25}$**  |  **$AP_{50}$** |  **$AP_{25}$**  | **$AP_{50}$**|
> |     VoteNet     |  22.64 |  9.04 |  24.93 | 12.82 |
> |  VoteNet + KNN  |  23.07 |  9.56 |  25.58 | 13.51 |
> | GroupFree[1] + KNN |  24.22 |  9.97 |  26.33 | 13.92 |
> |   3DETR[2] + KNN   |  24.08 | 10.21 |  26.01 | 14.36 |
> |       Ours      |  31.25 | 16.01 |  32.25 | 19.52 |
>
> [1] Liu, Ze, Zheng Zhang, Yue Cao, Han Hu, and Xin Tong. "Group-free 3d object detection via transformers." In Proceedings of the IEEE/CVF International Conference on Computer Vision (ICCV), 2021.
>
> [2] Misra, Ishan, Rohit Girdhar, and Armand Joulin. "An end-to-end transformer model for 3d object detection." In Proceedings of the IEEE/CVF International Conference on Computer Vision (ICCV), 2021.
>
>
> **Q3. Were other methods of updating the prototypes tried ?**
>
> **A3.** Thank you for pointing this out. Indeed, we included another method for updating the prototypes in the supplementary material. It calculates the similarity between a point feature with all geometric prototypes, and updates all geometric prototypes in a soft manner considering the similarity scores between a point feature and the geometric prototypes. The details can be found in Section A.3 in the supplementary material.
>
> **Q4. Does setting the prototype at the end (no updates) perform well ?**
>
> **A4.** As shown in the table below, for the proposed Prototypical VoteNet, if we don’t update the prototype in PVM, the performance would degrade significantly. Without updating, the randomly initialized prototypes can not learn the geometry information from base classes in the training phase. In this case, it is hard to transfer the basic geometry information from base classes to the novel classes as the prototypes are meaningless.
>
> |      | 3-shot |  3-shot     | 5-shot |   5-shot |
> |------------|:------:|:-----:|:------:|:-----:|
> | **Method**      |   **$AP_{25}$**  |  **$AP_{50}$** |  **$AP_{25}$**  | **$AP_{50}$**|
> | No updates |  28.05 | 13.89 |  28.51 | 14.51 |
> |   Updates  |  31.25 | 16.01 |  32.25 | 19.52 |

---

### Meta-Review · Area_Chair_oQWj · 2022-08-25

**Recommendation:** Accept
**Confidence:** Certain

**Metareview:**

The paper received mixed reviews. Two reviewers were fairly positive, on the basis of of the novelty of the problem, the quality of the results, the introduction of datasets and benchmarks, and the proposed method. Although the method combines two existing solutions to point cloud detection and few-shot, these reviewers considered that the paper shows that this combination is not trivial. They liked the implementation of the PVM and PHM modules as a means of disentangling the feature embedding and detection. The negative reviewers raised a number of concerns, including a diverging opinion that the combination of the two strategies is somewhat trivial and the paper lacks technical novelty, the fact that the datasets are mostly a combination of existing ones, that several baselines could have been borrowed from the 2D few shot literature for a more extensive evaluation, and other concerns of detail. The authors provided a very thorough rebuttal, which addressed many of these issues. In result, one of the negative reviewers mentioned that, despite the limitations, the paper is worth publishing and the other engaged in an extensive discussion with the authors, oscillating between positive and negative positions towards the paper at different points of the interaction. After discussion, there was a sense that no reviewer significantly opposed the publication of the paper. While the limitations above (somewhat limited technical and dataset novelty) hold, the novel nature of the problem, its potential interest for future work by the community, and the results achieved by the method were found to justify publication.

**Award:**

No

---

### Decision · Program_Chairs · 2022-09-14

Accept